# Combined Metabolome and Transcriptome Profiling Reveal Optimal Harvest Strategy Model Based on Different Production Purposes in Olive

**DOI:** 10.3390/foods10020360

**Published:** 2021-02-07

**Authors:** Guodong Rao, Jianguo Zhang, Xiaoxia Liu, Xue Li, Chenhe Wang

**Affiliations:** 1State Key Laboratory of Tree Genetics and Breeding, Research Institute of Forestry, Chinese Academy of Forestry, Beijing 100091, China; rgd@caf.ac.cn (G.R.); shiyoo02@163.com (X.L.); shiyoo01@163.com (X.L.); shiyoo03@163.com (C.W.); 2Collaborative Innovation Center of Sustainable Forestry in Southern China, Nanjing Forestry University, Nanjing 210037, China; 3Key Laboratory of Tree Breeding and Cultivation, National Forestry and Grassland Administration, Research Institute of Forestry, Chinese Academy of Forestry, Beijing 100091, China

**Keywords:** olive, fatty acids, polyphenols, hydroxytyrosol, oleuropein, harvest

## Abstract

Olive oil has been favored as high-quality edible oil because it contains balanced fatty acids (FAs) and high levels of minor components. The contents of FAs and minor components are variable in olive fruits of different color at harvest time, which render it difficult to determine the optimal harvest strategy for olive oil producing. Here, we combined metabolome, Pacbio Iso-seq, and Illumina RNA-seq transcriptome to investigate the association between metabolites and gene expression of olive fruits at harvest time. A total of 34 FAs, 12 minor components, and 181 other metabolites (including organic acids, polyols, amino acids, and sugars) were identified in this study. Moreover, we proposed optimal olive harvesting strategy models based on different production purposes. In addition, we used the combined Pacbio Iso-seq and Illumina RNA-seq gene expression data to identify genes related to the biosynthetic pathways of hydroxytyrosol and oleuropein. These data lay the foundation for future investigations of olive fruit metabolism and gene expression patterns, and provide a method to obtain olive harvesting strategies for different production purposes.

## 1. Introduction

Olive oil originated in the Mediterranean and is a widely recognized, high-quality, edible vegetable oil [1]. The consumption of olive oil has gradually increased over the past 10 years, even in some non-oil olive producing areas like Japan and Canada [2]. This increase in popularity is due in large part to its nutritional and health-promoting effects. Olive oil contains a high abundance of mono- and poly-unsaturated fatty acids that are more easily absorbed by humans [1,3]. Olive oil also contains many minor components that are beneficial to human health, including chlorophyll, polyphenols, and tocopherols. In particular, hydroxytyrosol, oleuropein, and squalene are substances in olive oil, which are otherwise very rare in plant oils and have great human health benefits owing to antioxidative capacity [4,5,6].

The quality of olive oils depends on many characteristics, including the specific cultivar, growth environments, and harvest times. Traditional harvesting of green fruits during the maturation period will reduce oil yields and increase the abundance of minor components, including polyphenols, while harvesting fully mature purple fruits will increase oil yields, but reduce polyphenol contents. Therefore, harvesting is typically conducted by mixing green fruits with semi-ripe or ripe fruits to ensure specific oil contents and polyphenol content characteristics [7,8]. Studies have examined the effects of harvest time on oil yields and qualities; however, studies on the effect of harvesting time on olive oil quality by comprehensive metabolite analyses in concert with gene expression analyses are rare [7,9]. The publication of the olive genome and the development of third-generation, full-length Iso-Seq sequencing technology has allowed more convenient and accurate investigation of olive polyphenol biosynthesis-related genes than was previously possible with second-generation Illumina RNA-seq sequencing technology alone [10,11,12].

In this study, we performed a comprehensive metabolomics analysis of three mature olive fruits (green fruit, semi-ripe fruit, and ripe fruit) collected at the same harvest time. Specifically, targeted analysis of 34 fatty acids (FAs) and 12 minor components was used in conjunction with targeted LC/MS and GC/MS metabolomics. Full-length Iso-Seq sequencing and Illumina RNA-seq sequencing were also conducted on the same fruit samples to evaluate gene expression. The metabolomic and full-length transcriptomic sequencing results were then used to propose a harvesting strategy model based on various considerations and the identification of genes related to hydroxytyrosol and oleuropein biosynthesis. These results provide a theoretical basis for subsequent genetic manipulation efforts and the development of improved olive cultivars.

## 2. Materials and Methods

### 2.1. Plant Material

Mature olive fruits including green, turning stage (semi-green/half-purple), and purple fruits (designated as F1, F2, and F3, respectively) were collected during the harvest season (15 October 2018) from 15-year-old olive trees (*O. europaea* L. cv. *Frantoio*) planted in experimental olive orchards at the Research Institute of Forestry of the Chinese Academy of Forestry in Longnan within the Gansu province. Samples were quickly frozen in liquid nitrogen and stored at −80 °C until further processing. Fruit pulps with six biological replicates were used for each sample in targeted metabolomics assays. Full-length Iso-Seq transcriptomics were conducted using a mixture of fruits at different maturity levels. Three biological replicates were used for each sample in Illumina RNA-seq transcriptomics.

### 2.2. GC/MS- and LC/MS-Based Targeted and Untargeted Metabolomic Analysis 

GC/MS-based fatty acid analysis and GC/MS-based targeted metabolomics analysis were conducted using previously described methods [13]. For LC/MS-based untargeted polyphenol analysis, we used standards to quantify the 12 minor nutrients. First, 100 mg of olive pulp was placed in a 5 mL centrifuge tube with five steel balls. The samples were then frozen in liquid nitrogen for 5 min, and then pulverized for 1 min by a sample preparation system (TISSUELYSER, Shanghai jingxin, China). One milliliter of methanol at −20 °C was then added and followed by vortexing for 30 s and sonication for 30 min at room temperature. Chloroform (750 μL) was then added along with 800 μL of hydrogen peroxide (at 4 °C) and followed by vortexing for 1 min. The mixture was then centrifuged at 12,000 RPM for 10 min at 4 °C, followed by a transfer of 1 mL of the supernatant to a new 1.5 mL centrifuge tube. The supernatant was then concentrated with a vacuum centrifuge concentrator and dissolved in 250 μL of a 2-chlorophenylalanine (4 ppm)/aqueous methanol solution (1:1, 4 °C). The sample was then filtered through a 0.22 μm membrane filter to obtain the sample for LC-MS analysis. Chromatographic separation was conducted on a Thermo Ultimate 3000 system equipped with an ACQUITY UPLC^®^ HSS T3 (150 × 2.1 mm, 1.8 μm, Waters) column maintained at 40 °C. The temperature of the autosampler was set at 8 °C. Gradient elution of the analytes was conducted with 0.1% formic acid in water (C) and 0.1% formic acid in acetonitrile (D) or 5 mM ammonium formate in water (A) and acetonitrile (B) at a flow rate of 0.25 mL/min. Injection of 2 μL of each sample was conducted after equilibration. An increasing linear gradient of solvent B (*v*/*v*) was used as follows: 0–1 min, 2% B/D; 1–9 min, 2–50% B/D; 9–12 min, 50–98% B/D; 12–13.5 min, 98% B/D; 13.5–14 min, 98–2% B/D; 14–20 min, 2% D-positive mode (14–17 min, 2% B-negative mode). LC/MS-based polyphenol measurements were conducted according to previous study [14].

### 2.3. RNA Extraction and Iso-Seq and RNA-Seq Sequencing

Total RNA was extracted from samples using a TIANGEN RNA Prep Pure Plant extraction kit (Tiangen Biotech Co. Ltd., Beijing, China) according to the manufacturer’s instructions. Degradation and contamination of RNA was evaluated by gel electrophoresis, and RNA purity was measured using a NanoPhotometer^®^ spectrophotometer (IMPLEN, Westlake Village, CA, USA). RNA concentration and integrity was measured using a Qubit^®^ RNA Assay Kit with a Qubit^®^ 2.0 Flurometer (Life Technologies, Carlsbad, CA, USA) and an RNA Nano 6000 Assay Kit with a Bioanalyzer 2100 system (Agilent Technologies, Santa Clara, CA, USA), respectively.

Pacbio Iso-seq cell library construction was conducted via first strand cDNA generation using the SMARTer PCR cDNA Synthesis Kit (Clontech, Tokyo, Japan) for reverse transcription of RNA. cDNA fragment integrity was evaluated using a BluePippin size selection system (Sage Science, Beverly, MA, USA), followed by cDNA normalization with a Trimmer-2 cDNA Normalization Kit (Evrogen, Moscow, Russia). Raw reads were processed to obtain error-corrected reads of inserts (ROIs) using the Iso-Seq pipeline. Full-length non-chimeric (FLNC) transcripts were then obtained by searching for polyA tail signals in addition to 5′ and 3′ cDNA primers in ROIs. ICE (Iterative Clustering for Error Correction) was used to obtain consensus isoforms and the FL consensus sequences were polished using Quiver. High-quality FL transcripts were then classified with the criterion of post-correction accuracy above 99%.

The NEBNext^®^ Ultra™ RNA Library Prep Kit for Illumina^®^ (NEB, Ipswich, MA, USA) was used for construction of Illumina RNA-Seq sequencing libraries. A total of 3 µg of RNA per sample was used as sequencing input material. Index oligonucleotides were added for sample sequence multiplexing as follows. Briefly, random hexamer primers and M-MuLV Reverse Transcriptase (RNase H-) were used for first strand cDNA synthesis, followed by second strand cDNA synthesis with DNA Polymerase I and RNase H. The remaining overhangs were then converted into blunt ends via exonuclease/polymerase activity. An Agilent 2100 Bioanalyzer and ABI StepOnePlus Real-Time PCR System were subsequently used for transcript purification, end preparation, and quality control of cDNA. The library was then sequenced on an Illumina HiSeqTM X10 platform. Raw reads were used to generate clean reads by removing reads with adapters, reads containing poly-N’s, and low-quality reads. All other downstream analyses were conducted with the high-quality clean reads. The clean reads were then mapped to the olive reference genome using the Tophat2 software program tools [10,15]. Only reads that were mapped perfectly or with one mismatch to the genome were included in further analyses and annotated based on the reference genome.

## 3. Results

### 3.1. Comprehensive Metabolomic Analysis of Olive Fruits at Different Maturity Stages

Targeted and untargeted metabolomic analyses were used to comprehensively investigate metabolites of olive fruits with different colors (F1–F3). In particular, targeted GC/MS-based analysis was used to investigate the fatty acid contents of olive fruits at different maturity stages, which were harvested during the same period (Figure 1A). A total of 34 named FAs were identified using several databases, including that of the National Institute of Standards and Technology (NIST), Wiley libraries, and that of an in-house database. Three dominant fatty acid components comprised over 95% of the total FAs in samples: oleic acid (C18:1Δ9c), palmitic acid (C16:0), and linoleic acid (C18:2Δ9c, Δ12c). The abundances of the three FAs increased with increasing maturity stages (Table 1).

The minor component abundances within the F1–F3 fruits were also investigated using targeted LC/MS metabolomics, yielding quantification of a total of 12 minor components such as polyphenols (Figure 1B) (Table 1). Greener, less mature fruits have been previously shown to contain higher polyphenol contents [4]. However, our analyses revealed a lack of uniformity among the polyphenols within all samples that were analyzed. Maslinic acid was generally the dominant polyphenol acid component in all samples, and its content exceeded 1000 μg/g in all samples. Squalene, rutin, and luteolin all exhibited moderate contents among fruit samples, with concentrations ranging between 50 and 300 μg/g. However, these three metabolites varied among the F1 to F3 samples. In particular, the F2 samples contained ~270 μg/g higher squalene contents than those of F1. However, the squalene content in the F3 samples was ~100 μg/g lower than in the F2 samples. In contrast, rutin exhibited relatively consistent concentrations among all three samples, while luteolin first decreased with increasing maturity and then increased. It is worth noting that tyrosol and hydroxytyrosol contents did not decrease with increasing fruit maturity, but rather significantly increased. Oleuropein exhibits antioxidant, anti-inflammatory, anti-atherogenic, anti-cancer, antimicrobial, antiviral, hypolipidemic, and hypoglycemic effects [16]. Oleuropein first exhibited increased concentrations with maturity stage, that later decreased (Figure 1B). Summing all of the FAs or polyphenols that were detected provides an approximation of the total content for these two metabolites within fruits at different maturity levels. The total fatty acid content increased notably during fruit maturation, but the total polyphenol content in F2 was lower than in the F1 and F3 samples (Figure 2A,B).

GC/MS- and LC/MS-based untargeted metabolomics analyses were used to obtain a more comprehensive evaluation of olive fruit metabolites at different maturity levels. A total of 181 metabolites were obtained and quantitatively analyzed. The metabolites were primarily classified as organic acids, polyols, amino acids, FAs, and sugars. Cluster and heat map analysis of the different metabolites and their respective abundances were then used to evaluate among-sample differences (Figure 2C,D, Appendix A). Eight phosphorylated compounds were identified and classified into two groups. One group comprised glucose 6-phosphate, fructose-6-phosphate, and sorbitol-6-phosphate, with highest contents in the F2 samples. The second group comprised fructose 1, 6-bisphosphate, pyrophosphate, phosphoric acid, monomethyl-phosphate, and myo-Inositol-1-phosphate, with F3 samples exhibiting the highest contents (Figure 2D). These phosphorylated compounds could also be produced by the methanol extraction procedure [17]. A total of 13 sugars were quantified among all samples and exhibited two different distribution patterns. One group exhibited highest contents in the lowest maturity stage fruits (green fruits) and gradual decreases with increasing maturity, suggesting that these sugars are consumed during maturation and may be used as an energy source during fruit development. The other metabolite group increased with fruit ripening and exhibited highest contents in the matured purple fruits, suggesting that these sugars may be essential metabolites in fruit ripening (Figure 2C). The polyol (Appendix A) and amino acid (Appendix A) contents were similarly distributed among fruits of different maturity stages, although purple fruit (F3) had the highest contents, while contents did not change between green and semi-green fruits (F1 and F2, respectively).

### 3.2. Transcriptomic Profiling with PacBio Full-Length and Illumina RNA-Seq

Olive fruits at different maturity stages were first mixed and then subjected to PacBio ISO sequencing. A total of 23.66 Gbp of clean data and 476,672 circular consensus (CCS) reads were obtained. Among these, 422,444 were full-length non-chimeric (FLNC) reads. The FLNC reads were clustered to obtain consensus sequences that were then polished to obtain a total of 43,748 high-quality consensus sequences. In addition, a total of 63.66 Gbp clean Illumina RNA-seq data were generated for nine samples (F1–F3 with three biological repeats), with the clean data (Q30 value greater than 91.09%) for each sample exceeding 6.44 Gbp (Appendix A). The second-generation transcriptomic data was then used to correct the Iso-Seq consensus sequences. After calibration and combination with high-quality consensus sequence for de-redundancy, a total of 19,958 transcript sequences were obtained.

The RNA-seq clean data from each sample were mapped to the reference olive genome [10], with mapping rates for all samples exceeding 91.56% (Appendix A). After combining the Iso-Seq and RNA-seq data, alternative splicing (AS) analysis was conducted on the whole gene dataset. Five major types of AS events were evaluated among transcripts [18]. A total of 202 AS events were observed; among these, intron retention was most prevalent (39.06%), followed by alternative 3′-splicing sites (31.88%), alternative 5′ splicing sites (14.26%), exon skipping (13.96%), and mutually exclusive exons (0.84%) (Figure 3A).

Long non-coding RNA (LncRNA) sequences were predicted using widely used modeling approaches including coding potential calculator (CPC) analysis, coding-non-coding index (CNCI) analysis, Pfam protein domain analysis, and coding potential assessment tool (CPAT) analysis [19,20,21,22]. A total of 281, 713, 267, and 999 LncRNAs were identified using the CNCI, CPAT, CPC, and Pfam methodologies, respectively. A total of 129 lncRNA transcripts were predicted using all four of the methods above (Figure 3B). The target genes corresponding to these 129 lncRNAs were predicted, and indicated that each lncRNA has several target genes (Appendix A).

The mapped reads were compared against the original genomic annotation of the reference olive genome to identify originally unannotated transcribed regions, identify new transcripts and genes of olive fruits, and supplement the original olive genome annotation. A total of 7998 new genes were identified in combination with new transcripts from AS sequences. The new genes were annotated using the COG, GO, Kyoto Encyclopedia of Genes and Genomes (KEGG), KOG, Pfam, Swissprot, eggnog, and Nr databases, resulting in the annotation of a total of 7962 new genes (Appendix A). The majority of annotated genes corresponded to known nucleotide sequences of plant species with 5073 (63.87%) transcripts sharing homology to those from Sesamum indicum, followed by Erythranthe guttata (599, 7.54%), and Coffea canephora (405, 5.01%) (Figure 3C).

To investigate differential expression of genes in fruits at different maturity stages, gene expression was normalized as fragments per kilobase of transcript per million (FPKM). In addition, Differential expression genes (DEGs) were detected based on criteria including a fold change of ≥2 and a false discovery rate (FDR) of <0.01. The green fruit (F1) and purple fruit (F3) samples exhibited the most DEGs, totaling 2820 DEGs with 1164 up-regulated and 1656 down-regulated. The F2_F3 comparison yielded the fewest DEGs (780) with 557 up-regulated and 223 down-regulated, while the F1_F2 comparison yielded the highest with 1204 DEGs comprising 417 that were up-regulated and 787 that were down-regulated (Figure 4A,B). The DEGs were annotated using the public databases described above, with 1189, 2782, and 773 DEGs annotated using the Nr database for the F1_F2, F1_F3, and F2_F3 comparisons, respectively (Appendix A).

Kyoto Encyclopedia of Genes and Genomes (KEGG) mapping analysis indicated that the DEGs primarily belonged to five pathways (Figure 4C). Most DEGs were ascribed to carbon metabolism-related pathways including those of photosynthesis, carbon fixation in photosynthesizers, carbon metabolism, and porphyrin and chlorophyll metabolism. The F1_F2 and F1_F3 comparisons had the most DEGs involved in the above-mentioned pathways, while the F2_F3 comparison had the fewest DEGs. Thus, these results suggest that the green fruits (F1) may require higher expression of genes involved in carbon biosynthesis for fruit ripening. A similar abundance of DEGs was observed in sugar metabolism-related pathways as were identified in the carbon metabolism-related pathways. These included the pentose phosphate pathway, pentose and glucoronate interconversions, glycolysis/gluconeogenesis, galactose metabolism, fructose and mannose metabolism, and starch and sucrose metabolism. Thus, green fruits may require higher expression of genes involved in energy metabolism in order to drive maturation. The most DEGs associated with fatty acid metabolism were enriched in the F1_F2 and F1_F3 comparisons, while the fewest DEGs were observed in the F2_F3 comparison group, indicating that genes associated with fatty acid biosynthesis reached stable levels in the half-green and purple fruits (F2). DEGs associated with flavonoid metabolism were mainly enriched in the F1_F2 comparison, which corresponded to samples that were green-colored, unripe fruits with high levels of polyphenol.

### 3.3. Combined Transcriptomic and Metabolomics Analysis of Hydroxytyrosol and Oleuropein Biosynthesis

Hydroxytyrosol can be present in the free form but is mostly present in the glycosylated form and esterified with elenolic acid, which forms oleuropein [1]. Moderate hydroxytyrosol concentrations (~20 μg/g) were measured in the fruits at different maturity stages. Hydroxytyrosol are scavengers of superoxide anions and hydroxyl radicals in addition to inhibitors of respiratory bursts of neutrophils and hypochlorous acid-derived radicals [23]. The hydroxytyrosol biosynthesis pathway contains nine metabolites and six primary genes, although a total of 178 genes are present in the pathway. These include 12 polyphenol oxidases (PPO), 4 tyrosine decarboxylases (TDC), 3 DOPA decarboxylases (DDC), 12 primary-amine (copper-containing) oxidases (CuAO), 3 phenylacetaldehyde reductases (PAR), and 144 alcohol dehydrogenases (ALDH) identified in the present study (Figure 5, Appendix A). Hydroxytyrosol contents were similar in green and semi-ripe fruits, but increased in mature fruits (Figure 1B). Several genes in the biosynthesis pathway exhibited similar trends, including 6 PPOs (OE6A026345, OE6A017966, OE6A058214, OE6A009108, OE6A063859, and OE6A046766), 1 DDC (OE6A082511), 1 CuAO (OE6A090395), 2 PARs (OE6A101520 and OE6A067462), and 16 ALDH(OE6A099552, OE6A039829, OE6A056721, OE6A014151, OE6A075679, OE6A115022, OE6A067462, OE6A101520, OE6A085091, OE6A082997, OE6A073145, OE6A033618, OE6A069825, OE6A042260, OE6A058913, and OE6A005696). These similarly expressed genes that correspond to hydroxytyrosol contents may play a key role in the biosynthesis of hydroxytyrosol and ripening of olive fruits.

Oleuropein is a hydroxytyrosol derivative belonging to the secoiridoid class that is abundant in the oleaceae family of plants. The biosynthetic steps leading to secoiridoid formation are still unclear because the genes are synthesized only in a restricted number of species, such as *Fraxinus excelsior*, *Syringa josikaea*, and *Catharanthus roseus* [24,25]. The complete biosynthetic pathway of oleuropein is still unclear; however, the pathway from geranyl diphosphate to secologanin has been elucidated, but the follow-up reactions are unclear. The known synthetic pathways include 12 metabolites and 9 associated gene families [3,26]. Here, we isolated and identified the putative genes involved in the oleuropein synthesis pathways. A total of 80 genes were identified in the three different olive fruit maturity stages including 3 geraniol synthases (GES), 11 geraniol 8-hydroxylases (G8H), 26 8-hydroxygeraniol oxidoreductases (8-HGO), 3 iridoid synthases (IS), 3 iridoid oxidases (IO), 18 7-deoxyloganetic acid-O-glucosyl transferases (7-DLGT), 9 7-deoxyloganic acid hydroxylases (7-DLH), 3 loganic acid methyltransferases (LAMT), and 4 secologanin synthases (SLS) (Figure 6). Oleuropein increased in the early maturation stages and then later decreased (Figure 1B). Some genes in the oleuropein biosynthetic pathway exhibited similar expression trends, including one GES (OE6A100919), two G8Hs (OE6A032847 and OE6A069232), five 8-HGOs (OE6A071931, OE6A071671, OE6A010935, OE6A019880, and OE6A089154), one IO (OE6A059843), two 7-DLGTs (OE6A106662 and OE6A066683), three 7-DLHs (OE6A073937, OE6A028145 and OE6A072427), and two LAMTs (OE6A091211 and OE6A111827). These trends and correlations with oleuropein contents indicate that these genes may be involved in oleuropein regulation.

## 4. Discussion

The optimal harvest time of olives is related to the cost of olive oil (oil yield) and oil quality (e.g., the distribution of unique metabolites in fruits including polyphenols). To obtain higher oil yields and higher qualities, the empirical strategy of harvesting that is generally used is to mix green fruits and semi-green fruits. Few producers use fully ripened fruits (e.g., purple fruits) to produce olive oil because fully ripened fruit is thought to contain very low polyphenols that are detrimental to olive oil quality [8,27]. The metabolomics data in this study indicated that total polyphenol contents did not significantly differ between green and purple fruits, and that the polyphenol contents of both fruit types were higher than those of semi-green fruits (Figure 2B). The combined use of green and semi-green fruits during harvesting can, indeed, achieve a high level of polyphenols. However, the total fatty acid content in purple fruits is significantly higher than in fruits that are not completely mature. Therefore, the empirical harvesting strategy indirectly increases the cost of producing olive oil.

The metabolomics data in this study indicated that the content of eight main fatty acids increased as the color became darker in olive fruits (Figure 1A), and the purple fruits indeed have the highest total contents of FAs (Figure 1B), suggesting that if producers want to increase the output of fatty acids, they should add more purple fruits, which is also consistent with the traditional harvesting method based on experience. The primary minor compounds hydroxytyrosol and oleuropein give extra-virgin olive oil its bitter, pungent taste. In addition, oleuropein exhibits pharmacological activities via antioxidant, anti-inflammatory, anti-atherogenic, anti-cancer, antimicrobial, antiviral, hypolipidemic, and hypoglycemic effects [16,28]. Oleuropein content increased from 15.18 μg/g in F1 to 22.32 μg/g in F2, and then decreased to 6.76 μg/g in F3, which had a similar or lower contents compared to previous studies [29,30,31] (Table 1). Therefore, if producers want to improve the quality of olive oil, from the perspective of total minor component content, green fruits and purple (ripe) fruits have the same effect. Thus, a good strategy to achieve the popular bitter and pungent tastes of olive oil is to extract oil from green and semi-green fruits rather than from ripe fruits. Squalene is a natural triterpene and one of the main components of skin surface lipids. It is a natural antioxidant and functions in skin hydration [32]. Squalene is also important in determining the quality of olive oil and can play a role in cancer prevention in humans [6,33,34]. The contents of squalene in green (F1), semi-green (F2), and purple fruits (F3) were 262.49 μg/g, 518.22 μg/g, and 425.14 μg/g, respectively. Thus, to obtain olive oil containing higher squalene abundances, an optimal harvesting strategy is to mix and harvest half-ripe and ripe fruits.

Generally speaking, different cultivars of olives would be planted in the same plantation so that the olives will mature in different periods, which can relieve the time pressure on harvest. Our research could help producers to determine the metabolites of different olive cultivars in the same plantation, and combine the actual production needs to give the best harvesting strategy. For example, we can mix cultivars with different maturity levels in order to achieve the best harvest ratio of olive fruits. Manual sorting of fruits of different colors will increase costs, but producers can use machines to sort fruits of different colors, which can greatly reduce labor costs. In addition to the pungent taste and bitterness mentioned above, olive oil also has pleasant flavors such as fruity, vegetable, and grassy flavors. Our research does have limitations in this regard. In the future, we can determine the source of these flavors by measuring aromatic or volatile metabolites, and further determine the best varieties and harvest ratios. Of course, our research only focuses on one olive cultivar, but we provide a set of harvesting strategy models, which will be of guiding significance for future research on the harvesting strategies of olive varieties.

Hydroxytyrosol and oleuropein are considered to be the most representative key nutrients in olive oil. In addition to giving olive oil special fragrance and taste, it can also promote human health. However, the biosynthetic pathway of hydroxytyrosol and oleuropein is unknown, and its biosynthesis mechanism cannot be analyzed at the molecular level, which is not conducive to promoting the transformation of the olive industry from traditional breeding to molecular breeding. This research used the combined Pacbio Iso-seq and Illumina RNA-seq gene expression data to identify genes related to the biosynthetic pathways of hydroxytyrosol and oleuropein. A total of 178 and 80 genes were identified in the hydroxytyrosol and oleuropein biosynthesis pathways, respectively. Although we have not fully analyzed the metabolic pathways of biosynthesis, our research can provide more abundant data for the subsequent study of these two biosynthetic pathways.

## 5. Conclusions

Here, we used combined metabolomics, full-length Iso-Seq and second-generation RNA-seq transcriptome approaches to investigate the relationships between metabolites and gene expression in olive fruits of different color that were collected during the same harvesting period. From these data, changes in fatty acid and polyphenol contents were measured in the fruits at different maturity stages, and a harvesting method was proposed that is optimal for different production purposes. Moreover, we identified putative genes related to the biosynthesis pathways of characteristic metabolites (i.e., hydroxytyrosol and oleuropein) in olive fruits. These comprehensive data and analyses lay the foundation for subsequent investigations of the relationship between metabolites and gene expression in olive fruits.

## Figures and Tables

**Figure 1 foods-10-00360-f001:**
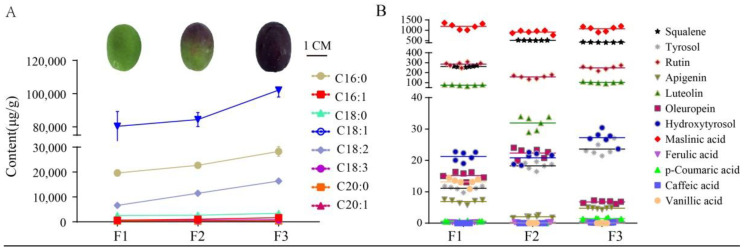
Targeted metabolomics analysis of olive fruits at different maturity stages. (**A**) Fatty acid contents determined by GC/MS-based targeted metabolomics. (**B**) Minor components (such as polyphenols) contents measured by LC/MS-based targeted metabolomics. “µg/g” for fresh weight; (**B**) has the same units on the y axis as (**A**).

**Figure 2 foods-10-00360-f002:**
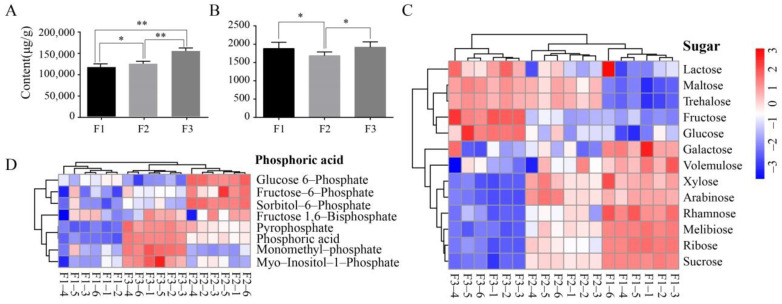
Metabolomics contents in olive fruits with different colors (F1–F3; green, turning stage (semi-green/half-purple), and purple fruits designated as F1, F2, and F3, respectively). (**A**) Total fatty acid contents among olive fruits at different maturity stages. (**B**) Total polyphenol contents among different mature olive fruits. Asterisks are significantly different (* *p* ≤ 0.05; ** *p* ≤ 0.01). (**C**,**D**) Heat map visualization of sugar and phosphorylated compounds.

**Figure 3 foods-10-00360-f003:**
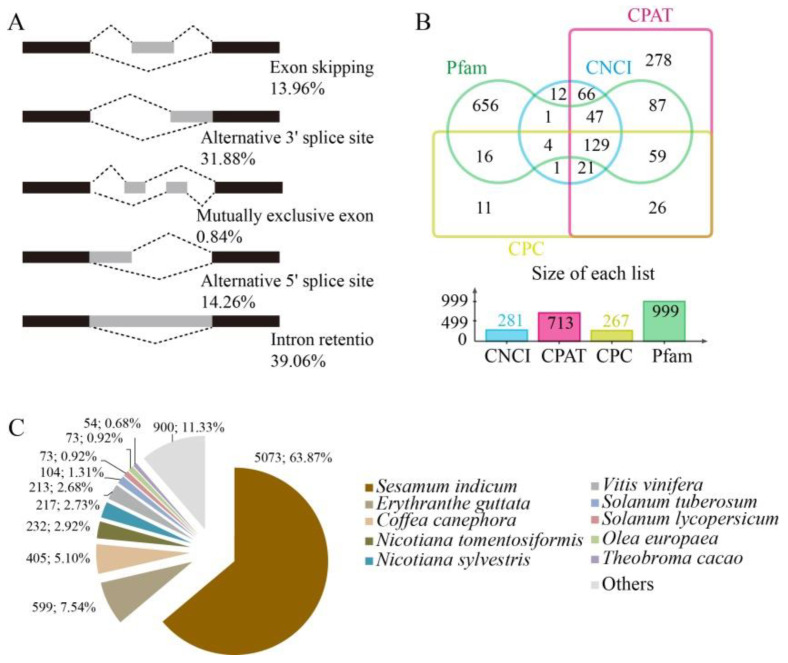
Transcriptomic based alternative splicing (AS), long non-coding RNA (LncRNA), and gene homologue analysis. (**A**) Five different major types of AS events were identified among transcripts. (**B**) Venn diagram showing LncRNAs identified by coding-non-coding index (CNCI), coding potential assessment tool (CPAT), coding potential calculator (CPC), and Pfam methods. (**C**) Distribution of homologous genes across species.

**Figure 4 foods-10-00360-f004:**
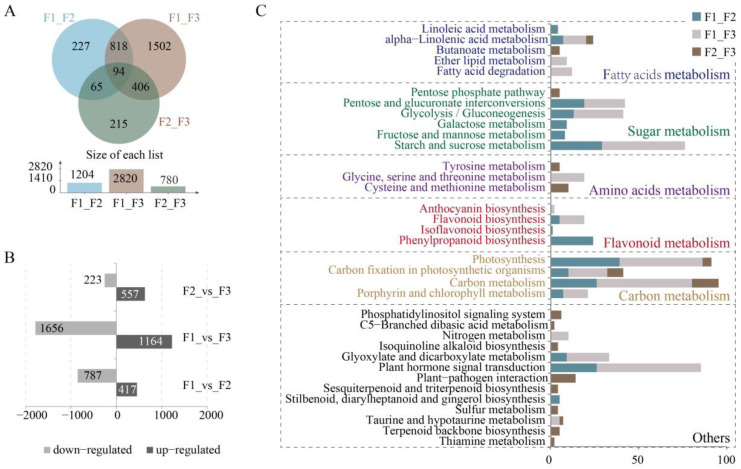
Transcriptomic-based gene identification and DEG analyses. (**A**) Venn diagram showing shared genes among each group. (**B**) Up- and down-regulated DEGs of each comparison group. (**C**) Kyoto Encyclopedia of Genes and Genomes (KEGG) classification of DEGs.

**Figure 5 foods-10-00360-f005:**
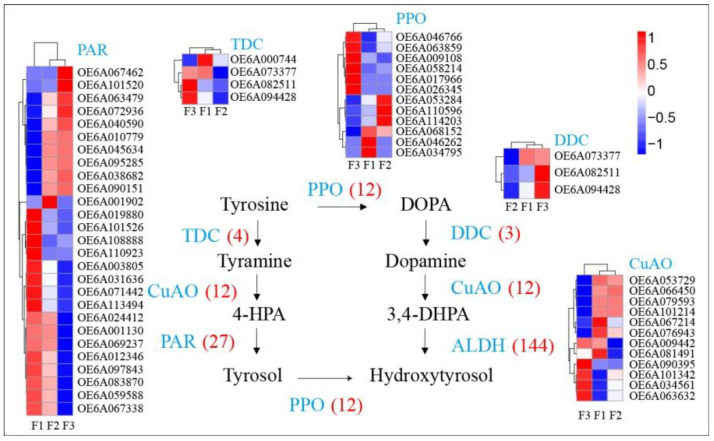
Expression of genes in hydroxytyrosol biosynthesis pathway and associated gene expression levels within olive fruits at different maturity stages.

**Figure 6 foods-10-00360-f006:**
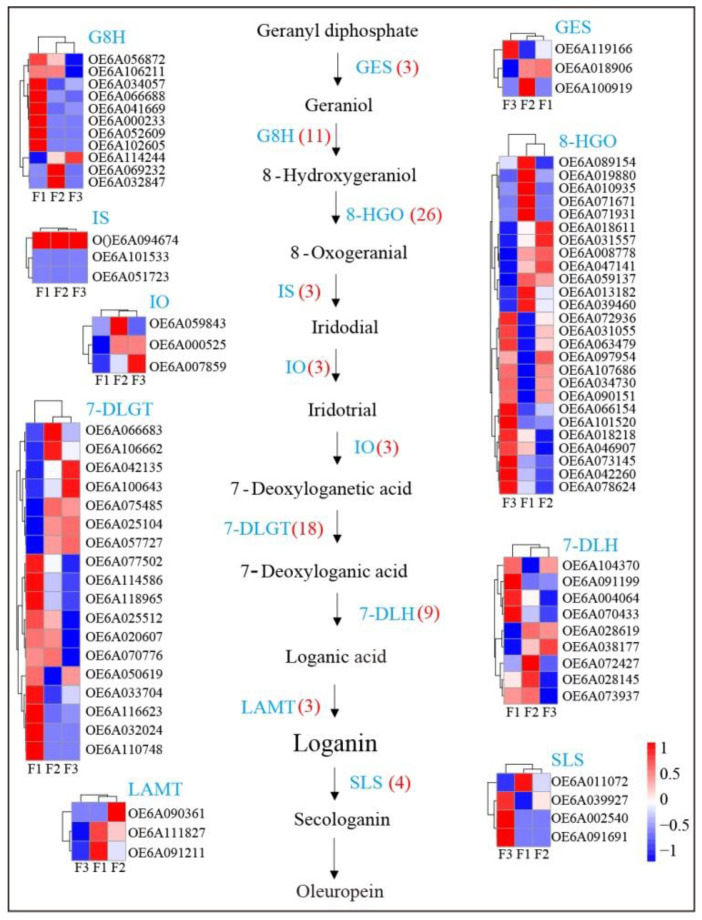
Expression of genes in oleuropein biosynthesis pathways and associated gene expression within olive fruits at different maturity stages.

**Table 1 foods-10-00360-t001:** Metabolite contents in fruits at different maturity stages determined by targeted metabolomics analyses. Six biological replicates were used to determine contents within groups. Data are shown as mean ± SD (μg/g). n.d., not detected.

Name	F1	F2	F3
C16:0	19,675.13 ± 1350.06	22,751.13 ± 801.68	28,291.71 ± 1905.85
C16:1	663.19 ± 67.49	1064.50 ± 67.88	1722.41 ± 97.58
C18:0	2587.96 ± 354.19	2661.12 ± 140.25	3453.95 ± 163.80
C18:1	80,340.02 ± 9798.77	84,364.30 ± 4234.48	102,126.63 ± 4138.54
C18:2	6612.13 ± 917.52	11,506.93 ± 487.85	16,425.71 ± 755.58
C18:3	755.37 ± 99.69	676.34 ± 55.74	833.11 ± 73.71
C20:0	354.63 ± 34.61	329.54 ± 20.78	391.51 ± 41.65
C20:1	355.15 ± 82.82	357.00 ± 58.59	412.05 ± 53.37
Vanillic acid	13.17 ± 1.22	n.d.	n.d.
Caffeic acid	n.d.	n.d.	0.21 ± 0.03
p-Coumaric acid	0.60 ± 0.08	n.d.	1.45 ± 0.16
Ferulic acid	0.27 ± 0.03	0.28 ± 0.05	0.61 ± 0.08
Maslinic acid	1189.25 ± 146.23	913.34 ± 86.91	1075.54 ± 122.00
Hydroxytyrosol	21.28 ± 1.54	20.83 ± 1.78	27.25 ± 2.22
Oleuropein	15.18 ± 1.26	22.32 ± 1.63	6.76 ± 0.41
Luteolin	74.64 ± 6.24	31.93 ± 2.22	102.21 ± 7.73
Apigenin	6.90 ± 0.70	2.05 ± 0.32	4.76 ± 0.29
Rutin	285.29 ± 21.91	157.00 ± 17.40	248.58 ± 20.18
Tyrosol	11.12 ± 0.77	18.28 ± 1.15	23.63 ± 2.05
Squalene	262.49 ± 6.40	518.22 ± 5.12	425.14 ± 7.42

## Data Availability

Sequence data generated in this study were deposited in GenBank under the bioproject accession numbers as follows: Pacbio Iso-seq data: SRR8692278; Illumina RNA-seq data: the three F1 biological repeats—SRR8690139, SRR8690138, and SRR8690141; the three F2 biological repeats—SRR8690140, SRR8690135, and SRR8690134; and the three F3 biological repeats—SRR8690137, SRR8690136, and SRR8690133.

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
