# Peer review of "Combined Metabolome and Transcriptome Profiling Reveal Optimal Harvest Strategy Model Based on Different Production Purposes in Olive"

_foods, 2021, doi:10.3390/foods10020360_

Round 1

Reviewer 1 Report

I’ve read with attention the paper of Rao et al. that is potentially of interest. The background and aim of the study have been clearly defined. The methodology applied is overall correct, the results are reliable and adequately discussed. I’ve only some minor comments:

  • The discussion is a bit short and unbalanced compared to the remain part of the text. So, it could be enriched
  • The authors should shortly discuss the potential limitation of their research approach
  • The text should be attentively revised before re-submission, because of some typos around the text.

Reviewer 2 Report

The manuscript by Guodong et al. describes a two pronged metabolomics and transcriptomics approach of studying phytochemicals in olive fruits. The manuscript is in general well written and the results are interesting, however the authors also need to address some issues in methodology and interpretation.

Line 19: Did you identify them or quantify them, if you claim identify you would need standards to be injected, otherwise if you use mass only how are you dealing with isobaric species

Line 35: Squalene is produced by humans, so the need to have it in the diet are debateable

Line 69/89: the degree of -80°C is the wrong character this happens elsewere in the manuscript too.

Line 69: Was the pitt removed or also pulverized?

Line 77-79: How were the samples pulverized exactly?

Line 95: Is model Mode?

Line 130-132: Is the cultivar you used the same as the one with the sequenced genome, otherwise the one mismatch might be too stringent.

Figure 1: is the µg/g for fresh or dry weight, has (b) the same units on the y axis as (a) [µg/g]? in the figure it is (a) in the Legend it is (A)--> Same for other figures. Squalene in (b) is not a polyphenol, and the others are phenolic compounds and not polypehnols as well, you should also check this in the rest of the manuscript. Polyphenols and phenolic compounds are not interchangeable!

Table 1: Does 0 mean it is not present or it was not detected n.d./n.a. or below limit of detection might be better here.

Line 181: Identified? How, where did you by standards from?

Line 185: Phosphoric acid related compounds? Some are phosphorylated sugars they originated from a sugar and ATP or other NTPs. A better term is phosphorylated compounds.

Line 188: This might be an artifact of your methanol extraction procedure see: ULLRICH J, CALVIN M. Mechanism of methyl phosphate formation by "killing" of spinach chloroplast fragments with methanol in the presence of phosphate. Biochim Biophys Acta. 1962 Feb 12;57:190-1. doi: 10.1016/0006-3002(62)91108-3. PMID: 13923696.

Line 221: How many events were observed in total and how are you destinguishing between immature mRNA and Exon retention?

Line 239: What does new mean, new for olive or fro plants in general?

Line 284: Are those genes really all involved in this biosynthesis pathway or are they related genes that now have a different function in the plant? It might be helpful here to explain how far the biosynthesis is already elucidated in olive.

Line 306: You only infered this you did not test for bioactivity of individual enzymes, so at best this is putative!

Line 307: Same as for the previous pathway! are they all involved in this or are they just closely related to genes that are. More explaination about what is knwon about the pathway would be great, especially if some work was done in olive. 

Line 328: Another reason for this is the way olives are mechanically harvested. How can you differentiate between the fruits at harvest? Also any selection and or hand picking of fruits would increase the cost of the produced oil significantly through the needed amount of labor and or machines. So to optimized for polyphenols would at best only be possible for premium products. Moreover, this analysis does not take into account any differences in taste that might exist between green and purple fruits that also determine the market value.

Line 340: In addition the fruits grow larger and contain more oil per fruit, so you get more oil in general if you wait for ripe fruits...

Line 368: They were at best putatativly identified! You did no experiments to charcterize the function of the gene/enzyme!
